# Investigation of Methods to Create Future Multimodal Emotional Data for Robot Interactions in Patients with Schizophrenia: A Case Study

**DOI:** 10.3390/healthcare10050848

**Published:** 2022-05-05

**Authors:** Kyoko Osaka, Kazuyuki Matsumoto, Toshiya Akiyama, Ryuichi Tanioka, Feni Betriana, Yueren Zhao, Yoshihiro Kai, Misao Miyagawa, Tetsuya Tanioka, Rozzano C. Locsin

**Affiliations:** 1Department of Psychiatric Nursing, Nursing Course of Kochi Medical School, Kochi University, Kochi 783-8505, Japan; 2Graduate School of Engineering, Tokushima University, Tokushima 770-8506, Japan; matumoto@is.tokushima-u.ac.jp; 3Graduate School of Health Sciences, Tokushima University, Tokushima 770-8509, Japan; aki.5.toshi.2@gmail.com (T.A.); taniokaryuichi@gmail.com (R.T.); fenibetriana@gmail.com (F.B.); 4Department of Psychiatry, School of Medicine, Fujita Health University, Toyoake 470-1192, Japan; zhao@fujita-hu.ac.jp; 5Department of Mechanical Engineering, Tokai University, Tokyo 259-1292, Japan; kai@keyaki.cc.u-tokai.ac.jp; 6Department of Nursing, Faculty of Health and Welfare, Tokushima Bunri University, Tokushima 770-8514, Japan; miyagawa@tks.bunri-u.ac.jp; 7Institute of Biomedical Sciences, Tokushima University, Tokushima 770-8509, Japan; tanioka.tetsuya@tokushima-u.ac.jp (T.T.); or locsin@tokushima-u.ac.jp (R.C.L.); 8Christine E Lynn College of Nursing, Florida Atlantic University, Boca Raton, FL 33431, USA

**Keywords:** schizophrenia, human–robot interaction, multimodal data, multimodal emotion recognition

## Abstract

Rapid progress in humanoid robot investigations offers possibilities for improving the competencies of people with social disorders, although this improvement of humanoid robots remains unexplored for schizophrenic people. Methods for creating future multimodal emotional data for robot interactions were studied in this case study of a 40-year-old male patient with disorganized schizophrenia without comorbidities. The qualitative data included heart rate variability (HRV), video-audio recordings, and field notes. HRV, Haar cascade classifier (HCC), and Empath API^©^ were evaluated during conversations between the patient and robot. Two expert nurses and one psychiatrist evaluated facial expressions. The research hypothesis questioned whether HRV, HCC, and Empath API^©^ are useful for creating future multimodal emotional data about robot–patient interactions. The HRV analysis showed persistent sympathetic dominance, matching the human–robot conversational situation. The result of HCC was in agreement with that of human observation, in the case of rough consensus. In the case of observed results disagreed upon by experts, the HCC result was also different. However, emotional assessments by experts using Empath API^©^ were also found to be inconsistent. We believe that with further investigation, a clearer identification of methods for multimodal emotional data for robot interactions can be achieved for patients with schizophrenia.

## 1. Introduction

Several developed countries are introducing robotic technology for geriatric nursing and healthcare practice, especially in high-tech healthcare environments for older patients with dementia [1] and patients with schizophrenia who have trouble perceiving or understanding the facial emotions of people with whom they communicate [2]. Furthermore, patients with dementia and schizophrenia have diminished facial expressivity, which is a common feature that interferes with effective interpersonal communication [2]. With rapidly developing humanoid robots, tremendous possibilities for investigating and improving the social competencies of people with social disabilities are anticipated. However, these possibilities of humanoid robots remain largely unexplored for patients with schizophrenia [3].

Cognitive dysfunction in schizophrenia should be conceptualized not as a linguistic disorder but rather as a communication disorder, which is the core clinical deficit among patients with schizophrenia [4]. Schizophrenia is characterized by the impairment of social cognitive processes [5], processing of negative verbal information [6], and emotional perception [7], all of which are necessary for daily communication. An impaired ability to process social information or to appropriately modulate interpersonal behavior, in particular, harms the social life of patients with schizophrenia [3,8].

Various studies have investigated human–robot interactions to improve the competencies of people with schizophrenia. Rus-Calafell et al. [9] found that patients with schizophrenia showed improvement in daily skills, social cognition, and performance after a virtual reality program. Another study by Cohen et al. [3] showed that in human–robot interaction, patients with schizophrenia face marked difficulties in exploiting facial social cues elicited by humanoid robots in modulating patients’ motor coordination. Moreover, Raffards et al. [10] concluded that humanoid robots have the potential to convey emotions to patients with schizophrenia.

Furthermore, a previous study [11] deployed the humanoid robot NAO [12] to determine the interpersonal coordination of patients with schizophrenia. The findings showed that non-intentional interpersonal coordination was preserved in patients with schizophrenia during human–robot interactions [11]. However, little is known about the interactions between patients with schizophrenia and artificially intelligent robots. Acquiring cognitive abilities such as detecting emotions from facial expressions and conversation contents necessitates relevant interactions between the robot and the patient. Enabling robots to accurately detect emotions while interacting with patients will enhance patient–robot interactions in expected ways, increasing the outcomes of robot function and use in healthcare settings. Japanese psychiatric hospital units intended for the long-term care of patients in the chronic phase face a shortage of healthcare staff, making available staff too busy to spend time with patients, develop therapeutic relationships, and implement psychosocial interventions [13]. Thus, healthcare robots must be able to effectively “behave” to assist nurses with patient care.

### 1.1. Aim

This study, as a case study, aimed to examine methods to create future multimodal emotional data for robots to interact with patients with schizophrenia.

### 1.2. Literature review of Heart Rate Variability, Facial Emotion Detection, and Speech Emotion Recognition

According to the relationship between heart rate variability (HRV) and emotion, low HRV and, subsequently, cardiac vagal control are associated with various negative emotions (anger, sadness, and fear) and maladies, including anxiety disorder, depression, and cardiovascular disease [14]. High trait anxiety was associated with reduced R–R intervals and high-frequency (HF) power across conditions [15]. Lower HF power is correlated with stress, panic, anxiety, or worry [16]. High trait anxiety is associated with reduced R–R intervals and HF power [15] and increased parasympathetic activity during the emotion regulation of situational empathic responses. Emotion regulation may be associated with changes in situational empathy and autonomic responses, preferentially dominated by the parasympathetic branch, possibly reflecting an increase in regulatory processes. Furthermore, this study provides evidence that empathy for different emotional valences is associated with distinct changes in situational empathy and autonomic responses [17].

Patients with schizophrenia have reduced facial activity; therefore, it has been reported as an emotion-specific impairment of positive facial expressions that has affected a deviant relation between facial emotional expressions and speech [18,19]. Several methods have been reported to detect facial emotions [20,21].

Riyantoko et al. [21] proposed a method that applies a Haar cascade classifier (HCC) and convolutional neural network (CNN) to recognize human facial expressions.

Abbaschian et al. [22] proposed solutions for the identified problems of speech emotion recognition (SER). They suggested creating a deep-learning-friendly database by combining datasets to compensate for the limited size of datasets. The use of transformers to build a language-aware model that adapts to the language to identify emotions for different accents was also proposed. Moreover, models that can classify speech emotions have been proposed to improve the robustness of SER models [22]. Xu et al. [23] applied multiscale area attention in a deep CNN to analyze emotional characteristics with varied granularities. Therefore, the classifier of SER can benefit from a combination of attention with different scales. Furthermore, Drakopoulos et al. [24] elaborated that deep learning algorithms are crucial for emotion discovery because the ability to extract non-trivial knowledge is more concise and structured compared to that of the original datasets. Furthermore, a survey conducted by El et al. [25] addressed three important aspects of designing a speech recognition system: the choice of suitable features for speech recognition, the design of the appropriate classification scheme, and the preparation of an appropriate emotional speech database for evaluating system performance.

Various surveys have been conducted to study methods that recognize emotions from speech using voice analysis [23,24,25,26,27]. Fahad et al. [26] demonstrated that automatic SER, based on digital signal processing and machine learning, can be used to enhance human–computer interaction systems. However, it does not achieve high accuracy in the natural environment because of the mismatch of the speaker, language used, text, and culture. Fahad et al. [26] reviewed survey papers of SER that described the emotion modeling methods, datasets, features, classifiers, mismatch environment, uncontrolled environment, and evaluation metrics in SER. In another survey, Reshma and Rajasree [27] proposed that the precision of SER depends on the database and a combination of features and classification. Features can differentiate between various emotional states using spectral features, such as the Mel frequency cepstral coefficient (MFCC) and Mel energy spectrum dynamic coefficient (MEDC); the classification framework sets speech data with appropriate emotions according to the selected features from speech [27]. It utilizes machine learning to gauge emotions based on features such as voice speed, intonation, and tone rather than semantics [28,29,30] and outputs the degree to which the five emotions of normality, anger, joy, sadness, and energy are expressed.

If human emotions can be inferred from these complex data, the accuracy of the robot’s emotion recognition may be improved. Rapid progress in humanoid robot investigations can potentially enhance the competencies of people with social disorders, although this enhancement of humanoid robots remains largely unexplored for people diagnosed with schizophrenia.

### 1.3. Research Hypothesis

As described above, HRV analysis is a method for analyzing human autonomic nervous system activity, and methodologies for estimating emotions such as stress reactions and empathy from this analysis are attracting attention. Among the methodologies that use artificial intelligence (AI), we used HCC and Empath API. In this study, we hypothesized that humanoid robots, using one or a combination of these methods to analyze human emotions, could recognize the emotions of patients with schizophrenia at the same level as that of humans. If this hypothesis holds true, incorporating useful emotion analysis methods into a conversational humanoid robot would enable the robot to recognize the emotions of a patient with schizophrenia at the same level as that of a human, select words according to changes in the patient’s emotions, and converse with the patient.

As a first step, we conducted a case study on a patient with schizophrenia to verify whether HRV, HCC, and Empath API^©^ could recognize the emotions of such a patient at the same level as that of a human by combining the above-mentioned analytical methods.

## 2. Materials and Methods

### 2.1. Design

This study used the intentional observational clinical research design (IOCRD) to simultaneously generate quantitative and qualitative data through observations and measurement processes using devices [31] (see Figure 1). Figure 2 shows the hardware and software used in this study.

The subject was a male patient in his 40s with disorganized schizophrenia (International Statistical Classification of Diseases and Related Health Problems, ICD-10, F20.1) and no comorbidities.

Data were collected in March 2021. Inclusion criteria: the subjects could converse in Japanese, did not have clinical psychiatric changes in their symptoms, such as those indicating schizophrenia, and were stable in their pharmacological treatment for schizophrenia and other diseases over the preceding three months.

Exclusion criteria: subjects under treatment for alcohol or drug abuse, aggressive or violent behavior, terminal illness, severe comorbidity, severe auditory impairment, inability to understand the Japanese language, or those unable to provide informed consent were excluded from this study.

### 2.2. Procedure for Data Collection

This study, using the application program for an intentional conversation, was jointly developed by Tanioka et al. and the Xing Company [32]. This program allows Pepper’s words and head movements to be remotely controlled by connecting it to two computer tablets (iPad minis) using a Wi-Fi connection. An operator and an assistant remotely input phrases into Pepper’s conversation application program via a tablet and keyboard from a separate room. The assistant aided the operator in inputting Pepper’s actions, such as acknowledgment gestures and nodding during the conversation. A subject conversed with Pepper. The researchers ensured that the subjects adhered to the study protocols during their conversation with Pepper, which lasted approximately 20–30 min.

This application was uploaded to the tablet using a keyboard through a Bluetooth connection to ensure the timely input of conversational texts. This allowed the operator to directly input the words for the conversation that they had in mind. This system also had a switch that made Pepper’s head and face move toward the person who was speaking. Additionally, this program included template responses, questions, and replies that fit the conversation.

The researcher led the patient to a chair and handed him a microphone after he was seated. The operator initiated the conversation by having Pepper greet the patient. The operator guided and controlled the subsequent conversation to ensure that the conversation proceeded well according to what the patient said and the situation. The conversation was about topics such as seasons and lifestyle. The conversation ended when Pepper made a remark that reminded us of the end of the conversation.

Data were collected in March 2021. This study used a radio clock to synchronize all data. Data include HRV, video and audio recordings (to record conversations and behavioral activities data during interaction with the robot), and field notes as qualitative data (to classify the interaction between the patient and robot). All the data were collected and verified.

Video recordings of conversations between the Pepper robot (Hereafter, Pepper) [33] and a patient were investigated and the sections where the patient’s facial expression changed were identified. This study was recorded by three digital cameras and two intermediary observers, who noted significant events occurring during the conversation.

### 2.3. Data Analysis Methods

For emotion expression analysis, conversations between the robot and patient, HRV analysis results, HCC, and acoustic analysis (voice) results were selected. This study analyzed four “time frame” results with subjective facial expression analysis by three expert healthcare professionals, HCC, and Empath API.

#### 2.3.1. HRV

The HRV of the patient was measured using an electrocardiogram to evaluate their sympathetic–parasympathetic balance (MemCalc/Bonaly Light; Suwa Trust, Tokyo, Japan) during their interaction with the robot. The coefficients of variation of R–R intervals and HF power on the electrocardiograph served as indices of parasympathetic nervous activity, while the low-frequency (LF)/HF ratio served as an index of sympathetic nervous activity. These data were analyzed at intervals of 2 s. Measurements of LF and HF power components were expressed in normalized units (LFnu, sympathetic nerve activity; HFnu, parasympathetic nerve activity).

#### 2.3.2. Subjective Facial Expression Analysis by Expert Healthcare Professionals

Two nurses (A and B) and a psychiatrist (C) evaluated the patient’s facial expressions. They watched a video-recorded conversation between the robot and the patient to understand the patient’s emotions during the conversation.

#### 2.3.3. Analysis by Facial Expression Recognition Algorithm Using HCC

Face recognition requires the detection of feature points of facial organs, such as the eyes, nose, and mouth. For the Haar-Like feature, the difference between light and dark is extracted by subtracting the sum of the pixel ground values of the black and white regions in the rectangular region to be calculated. Generally, multiple patterns of this rectangular region are prepared, and the obtained feature values of the differences between light and dark are combined to be used as the feature values for object detection.

HCC uses the facial expression recognition (FER) class [34], an open-source library for Python. The FER class accepts the parameter “mtcnn” as an argument, but the default parameter is “mtcnn = False”. The HCC included in the OpenCV library is used. FER can also analyze facial expressions in the video, but only the frames that successfully detect faces are output. FER can also perform facial expression analysis on the video, but it outputs only the analysis results for the frames in which face detection is successful. In this case, the correspondence between time and frames cannot be established, so this study divided the video into frames and analyzed the facial expressions in each frame.

It is an effective method for real-time object recognition because of its low computational complexity and high recognition speed. In this study, facial regions and organs were detected using the HCC [21], and then classified by a facial expression classification model (angry, disgust, fear, happy, sad, surprise, neutral).

#### 2.3.4. Sound Emotion Analysis

The Empath API^©^ (Empath) was used for acoustic analysis. The Empath is an emotion conversation AI that analyzes the quality and emotion of verbal conversations from audio files. By considering this, the value output from the Empath was doubled for suitable comparison with the other analyzed data and expressed as a ratio.

The Web Empath API, developed by Empath, Inc, is an API for analyzing emotions from voice data. Although there is a free version of the Web Empath API, it is not suitable for experiments due to the large limitation on the number of requests per month. Consequently, this study used the paid version of the Web Empath API, which has no upper limit on the number of requests. The specific analysis algorithm is not disclosed. Since speech sentiment analysis is performed in the cloud, it is not suitable for the batch analysis of long and continuous speech files. Specifically, the Web Empath API allows the transmission (POST) of voice files for less than 5 s per request. It utilizes machine learning to gauge emotions based on features such as voice speed, intonation, and tone, rather than semantics [28,29,30] and outputs the degree to which five emotions of normality, anger, joy, sadness, and energy are expressed on a scale of 0–50. If the audio file to be analyzed is longer than 5 s, it can be edited to less than 5 s by cutting off unnecessary sections before and after the audio file. The audio files were extracted frame by frame from the video (converted to the WAVE file format, at a sampling frequency of 11,025 Hz, and converted to mono) using ffmpeg-python, a library for handling video files that are available in Python. A single frame of an audio file lasts less than 5 s, so it is possible to analyze all audio files.

## 3. Results

The available data, HRV, subjective facial expressions, facial expression analysis, and utterance data amounted to only four times (Table 1). The HRV results of this patient showed a pulse rate of about 100 per min (HR-mean, 105.68 ± 2.3/min) during a conversation with Pepper and LFnu (80.84 ± 6.65%).

The result of Time A was happy as gauged by HCC, and calm as evaluated by Empath. HFnu, 28.525%; LFnu, 71.475%. Subjective facial expression analysis results by medical expert healthcare professionals: evaluator A, happiness and smile; evaluator B, laugh out loud; evaluator C, wry smile.

The result of Time B was recognized as “happy” by HCC and then calm and sorrow evaluated by Empath. HFnu, 22.932%; LFnu, 77.068%. Subjective facial expression analysis results by expert healthcare professionals: A, happiness; B, laugh out loud; C, wry smile.

The result of Time C was also happy in HCC and calm and sorrow in Empath. HFnu, 23.645%; LFnu, 76.355%. Subjective facial expression analysis results by expert healthcare professionals: evaluator A, happiness; evaluator B, laugh out loud; evaluator C, wry smile

The result of Time D was sad by HCC, calm, joy, and energy evaluated by Empath. HFnu, 8.726%; LFnu, 91.274%. Subjective facial expression analysis results by expert healthcare professionals: evaluator A, contempt; evaluator B, tilted his head little: Couldn’t he understand? Evaluator C tilted his head at Pepper’s surprising answer (People like winter because it’s white).

## 4. Discussion

### 4.1. Differences between Subjective Analysis by Expert Healthcare Professionals and HRV, HCC, and Empath

This study analyzed the results of subjective facial and voice expression analyses by three expert healthcare professionals, HRV, HCC, and Empath, in a patient with schizophrenia during conversations with robots. The patient did not verbally express emotions strongly but instead expressed positive or negative emotions through facial expressions.

The HRV analysis results showed a pulse rate of approximately 100 per min and an LFnu of approximately 80% during a conversation with Pepper. Sympathetic dominance persisted.

Negative symptoms of schizophrenia, such as affective blunting, avolition, and asociality, remain poorly understood [35]. Most of these symptoms have been evaluated on rating scales by the observer [36]. A previous study observed that reduced HRV was associated with higher disease severity, as assessed by the psychiatric symptoms [37]. There are many papers on schizophrenic patients’ impaired FER toward another person [38]. However, little is known about analyzing FER and SER based on machine learning algorithms of patients with schizophrenia. It has been reported that indicators of parasympathetic functioning, such as HF HRV and the root mean square of successive R–R interval differences (RMSSD), are reduced [39,40]. These indicate significantly lower vagal activity in individuals with schizophrenia than in healthy controls. It was reported that reduced vagal HRV, increased heart rates, and increased negative affect at rest were replicated when comparing a sufficiently large sample of patients with schizophrenia to healthy controls [41,42].

There is an increase in sympathetic tone when stressed [43]. Moreover, in the case of the patient who exhibited a relatively high heart rate, a conversation with the robot could have been stressful [44]. Notably, positive emotions such as affection, entertainment, and joy are associated with increased parasympathetic influence, and happiness is associated with increased sympathetic activation [45]. By contrast, negative emotions such as anger, anxiety, fear, or sadness are characterized by increased activation of the sympathetic nerves of the heart [17].

When the subjective facial expression analysis results by expert healthcare professionals, happiness, smile, or wry smile were evaluated, there were subtle differences in expressions and perceptions. In one scene, two expert healthcare professionals evaluated the patient’s facial expression as doubtful, but one regarded them as contempt. It is said that basic human facial expressions do not differ by race, but recent studies have reported that Japanese people are less likely to show some emotions [46]. In this study, positive facial expressions agreed with expert healthcare professionals’ evaluation. However, negative facial expressions were difficult to evaluate, similar to previous studies.

HCC and Empath generated different results in Time A: HCC reported “happy”. Empath estimated “calm”. From the results of Time B, it can be inferred that the voice is mostly devoid of positive emotions. By contrast, the results of the facial expression analysis by HCC were “happy”.

HCC recognized Time C’s result as “happy”. Empath estimated “calm” and “sorrow”.

In Time D, the result of facial expression analyses by HCC was “sad”. Empath reported “calm”. For subjective facial expression analyses, the results of each evaluation were different.

In Times A and B, the results of subjective analyses of facial expressions by evaluators A, B, and C were happiness and smile, laugh out loud, and wry smile, and happiness and smile, smile, and wry smile, respectively.

For Time C, the subjective facial expression analysis results of evaluators A, B, and C were happiness, laugh out loud, and wry smile, respectively.

Especially, all results of the evaluation were different in Time D. Because, Pepper said “I like winter because my body is white”. It was considered that the patient did not understand Pepper’s joke.

From the above, the human assessment results suggested that the patient was confused by Pepper’s answer. The calm values analyzed by Empath in Times A and D were high. Moreover, the values of happy analyzed by HCC in Times A and D were high. Comparing the utterances of the subjects and the subjective facial expressions and phonology analyzed by medical professionals, they did not match the data obtained by Empath.

It has been reported that patients with schizophrenia have auditory emotion recognition deficits such as the abnormal identification and interpretation of tonal or prosodic features that transmit emotional information in sounds or speech [47]. The effects of voice-synthesized phonology and tones used in robots on conversations between schizophrenic patients and robots are considered for future study.

The analysis results for voice and facial expressions are different because the classifier analyzes based on only one of them. Compared with expert healthcare professionals who observed and evaluated both voice and facial expressions, their differences by classifier may be reasonable. As a result of a comprehensive judgment by expert healthcare professionals on voice and facial expression, if there is no agreement between the subjects, it is considered that the data are difficult to judge, even by humans. Therefore, if the judgment results differ by the classifier only facial expression or voice, it is important to integrate multimodal to judge emotion.

### 4.2. Whether HRV, HCC, and Empath API^©^ Are Useful to Create Future Multimodal Emotional Data about Robot–Patient Interactions

It can be inferred from the HRV analysis that the heart rate is fast and that the sympathetic nervous system is in a significant state, as observed from the LFnu ratio. In this experiment, HRV data were obtained using a contact-type sensor. To mount the HCC on a robot in the future and obtain information, changing the method to obtain information using a non-contact-type sensor to obtain information on human stress, anxiety, and sense of security is required.

Facial expressions are the first-line visual data that can be sources of sophisticated sensory data inputted to enhance the understanding of the complex nature of variations in facial expressions.

During facial expression analysis, HCC evaluated negative emotions such as angry. An observable indicator of such emotions could be the use of silence or other conversational methods to refrain from speaking to the robot. Considering that the preference for facial expressions or voice to express emotions varies among individuals, it might be difficult to recognize emotions more accurately by just treating facial expressions and voices as features. Another method that could be implemented is to make the robot verbally confirm these hidden negative emotions. The HCC estimates human emotions from facial expressions, which are also considered similar to the evaluation results of human raters, while the results of the emotion analysis of speech by Empath were considered to differ from those of human evaluations.

The differences in emotion for each modal (HRV, HCC, Empath API) should create higher-quality, more accurate multimodal emotion data that can be used for learning robot–patient interactions. The differences in emotion estimation results between the modalities obtained in this experiment can be attributed to the lack of a unified learning model (each is built on different training data and has different emotion classes).

Thus, it is important to re-train (or fine-tune) using the same data source (audio and images in the video information) with publicly available machine learning algorithms. Furthermore, correct answer information (emotion labels) is necessary for re-training, so it is essential to create a certain amount of manual data. By constructing small-scale, high-quality training data on our own (or using publicly available data, if available) before creating large-scale training data, it is considered that researchers can “create multimodal emotion data” that can be used in actual conversations with patients. Therefore, further accumulation of multimodal data between the patient and robot is required in the future. In addition, if an explainable emotional AI can be developed, it would be useful in automatically constructing training data for multimodal emotional data.

Emotion analysis with HCC or Empath API does not provide a human-interpretable view of what criteria were used internally to make such judgments. In particular, complex neural network models, such as deep learning, are, in most cases, black boxes that render the basis of judgment impossible to understand. If it remains a black box, it will be difficult to improve and correct the trajectory when problems occur due to judgments made by AI.

Recently, research on Explainable AI has become active in various fields, such as computer vision, natural language processing, and pathology [48]. Google is developing the Explainable AI framework [49] that can show, for example, the meaning of AI output by visualizing it in a way that humans can understand. Moreover, some studies show how other workforce personnel, such as clinicians and physicians, used AI [50,51]. Datta et al. [50] envisioned a conversational approach to designing machine learning models to identify factors affecting the risk of hospital-acquired venous thromboembolism. Another study [51] used machine learning in order to find drug–drug interactions that can lead to adverse events. Furthermore, the results of this study highlight that AI has become more practical and useful, which is why it is being well considered in clinical settings. However, with the process of understanding human emotions yet to be elucidated, AI technology that can explain why such emotions occurred and were recognized has yet to emerge.

### 4.3. Limitations and Future Scope

As the results of this case study were derived from only one patient, we plan to increase the number of patients in future studies. However, considering the dearth of healthcare personnel in this field, the introduction of robots is highly beneficial to both patients and healthcare personnel. This study makes a significant contribution to the literature because it evaluates the benefits of the complementary operation of a combination of robotic classifier technology to analyze the facial expressions of patients with schizophrenia.

## 5. Conclusions

This case study examined methods to create future multimodal emotional data for robots to interact with a patient with schizophrenia. The research hypothesis questioned whether HRV, HCC, and Empath API^©^ are useful for creating future multimodal emotional data about robot–patient interactions. The evaluation result based on HCC was in agreement with the result of human observations in the case of rough consensus in human judgment. However, when human judgment disagreed, the HCC evaluation results also differed from the results of human judgment. Thus, when it is difficult for humans to make a judgment, it is considered that with AI, is also difficult to make a judgment. The HRV analysis showed that sympathetic dominance persisted, matching the conversation situation. The results of the expert healthcare professionals’ emotional assessment analyzed by Empath were inconsistent.

The differences in emotion for each model (HRV, HCC, Empath API) could create higher-quality, more accurate multimodal emotion data that can be used for learning robot–patient interactions. The differences in emotion estimation results between the modalities obtained in this experiment could be attributed to the lack of a unified learning model. It was considered important to re-train (or fine-tune) using the same data source using publicly available machine learning algorithms.

## Figures and Tables

**Figure 1 healthcare-10-00848-f001:**
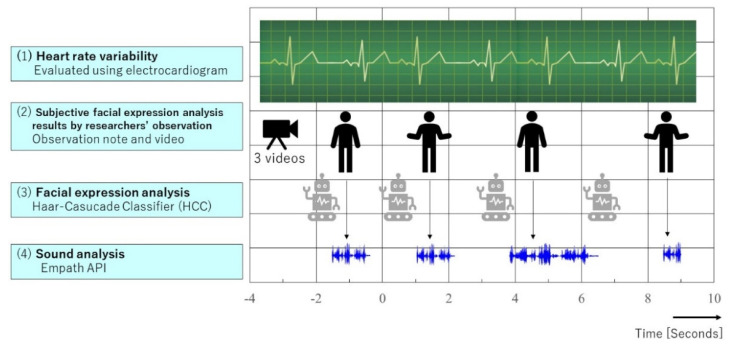
Research framework by IOCRD.

**Figure 2 healthcare-10-00848-f002:**
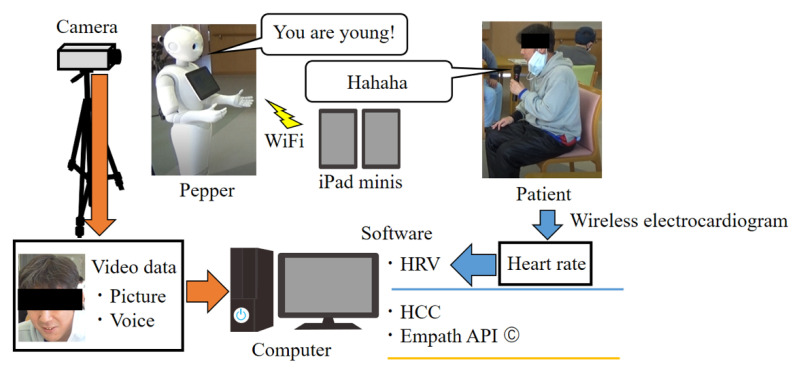
Hardware and software used in this study.

**Table 1 healthcare-10-00848-t001:** Results of the available data, HRV, subjective facial expressions, facial expression analysis, and utterance data.

Time	Time	A	B	C	D
	Elapsed Time	9:33:06	9:35:48	9:36:58	9:41:12
**Heart rate variability**	HR-mean	102.6	106.5	105.1	109.6
	HF	11.357	4.803	4.093	3.001
	LF	28.461	16.143	13.216	31.39
	HFnu	28.525	22.932	23.645	8.726
	LFnu	71.475	77.068	76.355	91.724
**Subjective facial expressions**	Evaluator A	Happiness and smile	Happiness and smiles	Happiness	Contempt
	Evaluator B	Laugh out loud	Smile	Laugh out loud	Tilted his head a little. (Couldn’t he understand?)
	Evaluator C	Wry smile	It seemed that he was surprised when asked “what shampoo. Wry smile	Wry smile	He tilted his head at Pepper’s surprising answer (Pepper likes winter because it’s white).
**Facial expression analysis**	HCC	Angry 0.09, disgust 0, fear 0.18, happy 0.58, sad 0.08, surprise 0.01, neutral 0.04	Angry 0.01, disgust 0, fear 0.01, happy 0.94, sad 0, surprise 0.03, neutral 0.01	Angry 0.01, disgust 0, fear 0.01, happy 0.86, sad 0, surprise 0.06, neutral 0.06	Angry 0.07, disgust 0, fear 0.185, happy 0, sad 0.67, surprise 0, neutral 0.086
**Conversation content**	Pepper	You are young			
	Patient	Hahaha	Ummm	Ha, once more	Ummm, white…, ok Ummm, that’s why white…, ok
**Sound analysis**	Empath API^©^	Calm: 1.00, anger: 0, joy: 0, sorrow: 0, energy: 0	Calm: 0.44, anger: 0.18, joy: 0.02, sorrow: 0.32, energy: 0	Calm: 0.84, anger: 0, joy: 0, sorrow: 0.14, energy: 0	Calm: 0.35, anger: 0, joy: 0.28, sorrow: 0, energy: 0.28

## Data Availability

The data presented in this study are available on request to the corresponding author. The data are not publicly available owing to privacy and ethical restrictions.

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
