# Peer review of "Investigation of Methods to Create Future Multimodal Emotional Data for Robot Interactions in Patients with Schizophrenia: A Case Study"

_healthcare, 2022, doi:10.3390/healthcare10050848_

Round 1

Reviewer 1 Report

  • The authors researched an interesting and novel topic involving human-robot interaction toward improving the competencies of people with schizophrenia. As the paper noted, the results were derived from only one patient limiting the work's generalizability. However, the most critical problem of the paper is that the article is only reporting raw data. There is no significant discussion on the results. This is partly because the authors did not construct a concrete research question based on the current literature. There is no sufficient integration of the literature, resulting in no hypothesis.

  • As the paper noted (Page 2 1.1. Aim), the study's goal is to examine the methods to generate data for HRI with schizophrenia patients. Just examining methods should remain as a tech report. One can not publish a scientific paper without a thorough analysis of the data and the investigation of a hypothesis.

  • The methods should be more detailed. The article stated that there was a conversation between the patient and robot (Pepper) but did not state what kind of conversation. What was the given theme? Was it a light social conversation? How was the patient guided in the study?

  • The discussion does not read as a discussion but rather as an extended results section. The discussion does not advance the scientific knowledge involving HRI with schizophrenia patients.

  • The paper needs to be proofread by a native speaker. 

Reviewer 2 Report

The manuscript discusses one of the interesting topics. The overall manuscript is structured and organized well. 

 Changes:

  1. I suggest, if possible including comparison results with existing/related works.
  2. I suggest including software and hardware setup sections. 
  3. In the current study, how many subjects were used for the data collection, and please include the details of the subjects and prior knowledge of the system?
  4. Please extend the conclusion section. 

Reviewer 3 Report

In this work, Osaka et al. investigates methods to create multimodal data for robot interactions in clinical settings. The manuscript is mostly written well. I have listed my comments/concerns below.

  1. Have you looked at explainability in AI based systems? It is important to highlight the need of oversight, interpretability and incorporating domain experts in the space.  Some papers that you should cite. Please find some more articles in this space in your related work section.

Lötsch, J.; Kringel, D.; Ultsch, A. Explainable Artificial Intelligence (XAI) in Biomedicine: Making AI Decisions Trustworthy for Physicians and Patients. BioMedInformatics 2022, 2, 1-17. https://doi.org/10.3390/biomedinformatics2010001

Since the authors are considering robotics in clinical settings, you should add a few pointers as to how the workforce like clinicians and doctors can be use for explainable AI. Some related works in this space are domain experts/clinician guided studies:

Datta A, Matlock MK, Le Dang N, et al. 'Black Box' to 'Conversational' Machine Learning: Ondansetron Reduces Risk of Hospital-Acquired Venous Thromboembolism. IEEE Journal of Biomedical and Health Informatics. 2021 Jun;25(6):2204-2214. DOI: 10.1109/jbhi.2020.3033405. PMID: 33095721.

Another work looks at interpretable ML to identify drug-drug interactions from EHR data by engaging with experts/workforce from the field 
Datta A, Flynn NR, Barnette DA, Woeltje KF, Miller GP, et al. (2021) Machine learning liver-injuring drug interactions with non-steroidal anti-inflammatory drugs (NSAIDs) from a retrospective electronic health record (EHR) cohort. PLOS Computational Biology 17(7): e1009053. https://doi.org/10.1371/journal.pcbi.1009053

  Explainability and Interpretability is key in this space. As you put more experts in the process, it is important to highlight what they bring to the table that AI cant.  

2. Have you looked at calibration of these classification models???

3.The overall analysis looks good.

4. How do you plan to use the multimodal emotional data for training robots? 

Round 2

Reviewer 1 Report

I have read the revised manuscript carefully. The paper improved vastly with appropriate responses to my feedback.

Reviewer 3 Report

The authors have addressed my comments adequately.